# Changes in AmotL2 Expression in Cells of the Human Enteral Nervous System in Oxaliplatin-Induced Enteric Neuropathy

**DOI:** 10.3390/biomedicines12091952

**Published:** 2024-08-26

**Authors:** Rebeca González-Fernández, Rita Martín-Ramírez, María-del-Carmen Maeso, Alberto Lázaro, Julio Ávila, Pablo Martín-Vasallo, Manuel Morales

**Affiliations:** 1Laboratorio de Biología del Desarrollo, UD de Bioquímica y Biología Molecular, Universidad de La Laguna, Av. Astrofísico Sánchez s/n, 38206 San Cristóbal de La Laguna, Spain; refernan@ull.edu.es (R.G.-F.); rmartira@ull.edu.es (R.M.-R.); javila@ull.edu.es (J.Á.); 2Instituto de Tecnologías Biomédicas, Universidad de La Laguna, C/Sta. María de la Soledad, Sección Medicina, 38071 San Cristóbal de La Laguna, Spain; 3Servicio de Patología, Hospital Universitario Nuestra Señora de la Candelaria, 38010 Santa Cruz de Tenerife, Spain; 4Laboratorio de Fisiopatología Renal, Departamento de Nefrología, Instituto de Investigación Sanitaria Gregorio Marañón, Hospital General Universitario Gregorio Marañón, 28007 Madrid, Spain; alberlaz@ucm.es; 5Departamento de Fisiología, Facultad de Medicina, Universidad Complutense de Madrid, 28040 Madrid, Spain; 6Servicio de Oncología Médica, Hospital Universitario Nuestra Señora de Candelaria, 38010 Santa Cruz de Tenerife, Spain

**Keywords:** AmotL2, enteral nervous system, oxaliplatin toxicity, gastrointestinal chemotherapy toxicity, enteral nervous system toxicity

## Abstract

Gastrointestinal (GI) toxicity is a common side effect in patients undergoing oxaliplatin (OxPt)-based chemotherapy for colorectal cancer (CRC). Frequently, this complication persists in the long term and could affect the efficacy of the treatment and the patient’s life quality. This long-term GI toxicity is thought to be related to OxPt-induced enteral neuropathy. AmotL2 is a member of the Angiomotin family of proteins, which play a role in cell survival, neurite outgrowth, synaptic maturation, oxidative stress protection, and inflammation. In order to assess the role of AmotL2 in OxPt-induced enteral neuropathy, we studied the expression of AmotL2 in cells of the enteric nervous system (ENS) of untreated and OxPt-treated CRC patients and its relationship with inflammation, using immunofluorescence confocal microscopy. Our results in human samples show that the total number of neurons and glial cells decreased in OxPt-treated patients, and TNF-α and AmotL2 expression was increased and colocalized in both neurons and glia. AmotL2 differential expression between OxPt-treated and untreated CRC patients shows the involvement of this scaffold protein in the inflammatory component and toxicity by OxPt in the ENS.

## 1. Introduction

Colorectal carcinoma (CRC) is the second most common malignant neoplasm in men and the third in women worldwide [1]. Although a high cure rate might be achieved through early diagnosis, many patients are diagnosed in an advanced stage of disease, either local or metastatic. In those cases, an increase in life expectancy and even a cure can be achieved by using combined treatments that include surgery, chemotherapy, and radiation therapy [1,2]. The combination of 5-fluorouracil (5-FU), folinic acid, and oxaliplatin (OxPt), FOLFOX, is the chemotherapy of choice for the adjuvant or neoadjuvant therapy of colorectal carcinoma [1].

Nausea, vomiting, constipation, and diarrhea are common gastrointestinal (GI) complications for OxPt-based chemotherapy, which might cause inadequate treatment administration or compromise the patient’s quality of life [3]. It is established that GI toxicity derives from mucositis produced by 5-FU and OxPt [4,5]. Mucositis is characterized by the presence of inflammation, ulceration, alteration of the intestinal flora, and GI secretion [6]. Elevated cytokines such as interleukin 1b and tumor necrosis factor (TNF)-α have been found in mucositis related to higher inflammation status [7]. Furthermore, GI symptoms might persist from months to 10 years after completion of treatment even though mucosa recovers, suggesting that chemotherapy might also damage the enteric nervous system (ENS), a complex network of neural circuits critical for regulating gastrointestinal functions like motility or mucosal secretions [6,8]. In murine models, it has been shown that chronic treatment with cisplatin and OxPt is associated with alterations in intestinal motility and a significant loss of neurons, adrenergic and sensory fibers, and glial cells in the ENS [9]. To our knowledge, this fact has not been shown so far in humans.

Scaffoldins are a group of proteins that facilitate macromolecular interactions and integrate the activity of various cellular functions such as motility, secretion, signaling, polarity, and angiogenesis [10,11,12,13]. Previous studies of our group found a differential gene expression pattern for scaffold proteins in peripheral leucocytes from patients with colorectal carcinoma treated with OxPt-based chemotherapy. Specifically, AmotL2 (Angiomotin-like 2) was one of the most affected [13]. Furthermore, OxPt-based chemotherapy produces significant changes in AmotL2 expression among the different cell types in healthy and malignant tissues from CRC cancer patients, showing the greatest variability in cells of the immune system and of the myenteric plexus [14].

The Angiomotin family of proteins comprises three closely related scaffold proteins Angiomotin (Amot), Angiomotin-like 1 (Amotl1), and Angiomotin-like 2 (Amotl2). Angiomotins play an important role in the regulation of cellular adhesion, cell polarity, and migration [15] by modulating Hippo cellular signaling pathways [16]. A dual role, often contradictory, of the Motin proteins functioning as oncogenes or tumor suppressors has been suggested in tumorigenesis [17]. Recently, an important role of Amot in the central nervous system has been shown [18]. Amot controls the organization of the postsynaptic machinery in the brain through the regulation of actin in dendritic spines and interaction with some synaptic proteins [19] and has been identified as a novel critical mediator of dendritic morphogenesis in cultured hippocampal neurons and Purkinje cells in vivo [20]. Amotexpression increases during neural differentiation and regulates YAP localization during differentiation. Amot leads Amot-dependent regulation of YAP in a human pluripotent stem cell’s fate [21].

The research and literature are very limited and neither the expression nor functions of Amotl2 in the ENS, neurons, or glia have never been investigated.

Given the role of AmotL2 in cell survival, neurite outgrowth, synaptic maturation, protection from oxidative stress, inflammation, and organ growth [15,22], the goal of this study was to find the involvement of AmotL2 in OxPt-induced myenteric plexus inflammation and damage.

The ENS is formed mainly by neurons (whose bodies form enteric ganglia), enteric glial cells, neural connections between ganglia, and effector nerves fibers that regulate muscle contraction, blood supply, and gastrointestinal and pancreatic secretion [23]. To study molecular variations in this functional and morphological complex structure, the use of regular methods, such as Western blot or PCR techniques, do not allow us to know specifically what the damaged cells are; instead, we decided to use confocal microscopy with well-characterized specific antibodies.

We found in human samples significant alterations in the number of neurons and glia cells after OxPt-based chemotherapy and a differential expression of AmotL2 related to that of TNF-α, indicating that this scaffold protein could have a role in the pathophysiology of OxPt-induced enteric neuropathy.

## 2. Materials and Methods

### 2.1. Patients and Samples

This study was approved by the Ethics Committee of La Laguna University, Canary Islands, Spain, and the Ethics Committee of the Hospital Universitario Nuestra Señora de Candelaria (HUNSC) CEIBA2016-0221 from 23 November 2016. A total of fifteen human samples were studied in three groups: a group of six patients diagnosed with locally advanced or metastatic CRC were selected, who received OxPt-based neoadjuvant chemotherapy and subsequently underwent the resection of the primary tumor. A second group of six patients with CRC who underwent surgery prior to OxPt-based chemotherapy and whose previously mentioned variables were collected. In both groups, the samples for the study were obtained from the end of the surgical resection piece with the greatest distance to the tumor. In the third group, normal controls, samples from three men were obtained from resections after traffic trauma from otherwise healthy patients. Exclusion criteria were previous treatment with chemotherapy, radiation therapy, or corticosteroids (with the exception of those used in antiemetic treatment). The following variables were collected from each patient: sex, age, comorbidities, concomitant medications, pathological stage, number of chemotherapy cycles administered before surgery, and the grades of neurological and digestive toxicity (according to the last version of the National Cancer Institute’s Common Terminology Criteria for Adverse Events). The paraffin-embedded samples of the selected patients were obtained from the Service of Pathology.

### 2.2. Immunohistochemistry

Paraffin-embedded tissue sections deparaffinized in xylene and hydrated in a graded series of alcohol baths were used. Epitope retrieval was performed by heating samples at 120 °C for 10 min in sodium citrate buffer (pH 6.0) using an autoclave. After blocking with 5% bovine serum albumin in Tris-buffered saline (TBS) for 1 h at room temperature, double immunofluorescence/simultaneous staining were performed, incubating tissue sections overnight at 4 °C simultaneously with mouse monoclonal anti-Glial Fibrillar Acidic Protein (GFAP) (dilution 1:100; #G3896 Sigma, Saint Louis, MO, USA), goat polyclonal anti-GFAP (dilution 1:500; #SAB2500462 Sigma, Saint Louis, MO, USA), anti-Microtubule-Associated Protein-2 (MAP2) (dilution 1:500; #MAB378 Chemicon International, Temecula, CA, USA), mouse monoclonal anti-TNF-α (dilution 1:150; #sc-52746; Santa Cruz Biotechnology, Dallas, TX, USA), and rabbit polyclonal TNF-α (dilution 1:75; # SAB4502982 Sigma, Saint Louis, MO, USA) in combination with rabbit polyclonal antibody against AmotL2 (dilution 1:50; #LS-C178611; LifeSpan BioSciences, Seattle, WA, USA). Negative controls were performed following the same protocol but without primary antibody. After washing three times with TBS, slides were incubated at room temperature for 1 h in dark with a mixture of two or three secondary antibodies: Fluorescein isothiocyanate (FITC)-conjugated goat polyclonal antibody against rabbit IgG (dilution 1:200; #F9887; Sigma-Aldrich, Saint Louis, MO, USA), goat polyclonal antibody against mouse IgG DyLight 650 (dilution 1:100; #ab97018; Abcam, Cambridge, UK), and Cy3 Affinity Pure Donkey Anti-goat IgG (H + L) (dilution 1:400; #705-165-147; Jackson Immunoresearch, Cambridge, UK). Finally, slides were mounted with ProLong^®^Diamond Anti-fade Mountant with DAPI (Molecular Probes by Life Technologies, Carlsbad, CA, USA) and analyzed using Leica SP8 confocal microscope (Leica Microsystems, Wetzlar, Germany).

### 2.3. Image Quantitative Analysis and Statistics

Image analysis was carried out using ImageJ software, https://imagej.net/ij/, accesed on 21 June 2024 (National Institutes of Health; Bethesda, MD, USA) with the EzColocalization plug-in [24]. Confocal images used for analysis were taken using the same parameters. Laser power and detector gain settings were optimized to cover fluorescence signals in a 16-bit depth range without saturation. Changes in fluorescence from baseline were measured as mean intensity of selected regions of interest. Complementarily, the “Cell Counter” plug-in was used to ensure that neurons or glia were counted only once. For statistical analysis, SPSS version 25 for Windows (IBM Corp., Armonk, NY, USA) was used. A dependence test (chi-square) was performed between the staining levels of each group of cells, and the non-parametric Kruskall–Wallis test was used to analyze significant differences in the distribution of staining levels with respect to cell type. *p* < 0.05 was considered statistically significant.

## 3. Results

### 3.1. ENS Neurons and Glial Identification

As the first step, we checked neurons and glial populations in ENS by fluorescence immunohistochemistry and confocal microscopy with MAP2 and GFAP markers in healthy colon areas from untreated and OxPt-treated CRC patients (Figure 1). Strikingly, some cells co-stained for both MAP2 and GFAP markers were found in samples from untreated and OxPt-treated CRC patients.

### 3.2. Quantification of Neurons and Glial Cells of the ENS

Cells from a total of five 20× myenteric plexus of confocal microscopy preparations of samples from non-treated and from pre-surgery-treated CRC patients were counted. The same surface area was delimited for counting in both conditions. A total of 2500 cells were counted. Figure 2 depicts the percentage of neurons and glial cells out of the total cells counted. The total number of neurons (MAP2 staining) and glial cells (GFAP staining) was estimated as a percentage of the total number of cells (DAPI staining).

### 3.3. TNF-α Expression in the ENS

The inflammation state of neurons and glia was evaluated by fluorescence immunostaining for TNF-α and confocal microscopy analysis, along with cells markers MAP2 and GFAP (Figure 3 and Figure 4). TNF-α-positive staining at low intensity level was found in healthy colon ENS areas from untreated CRC patients, mainly in GFAP+ glial cells, while MAP2+ neurons showed much lower or absent signals (Figure 3). In samples from OxPt-treated patients (Figure 4) the TNF-α signal was more intense, varying from medium to high levels, mainly in GFAP+ cells.

### 3.4. OxPt Treatment Effects over AmotL2 Expression

To check AmotL2 involvement in inflammation and toxicity elicited by OxPt-based chemotherapy in the ENS, AmotL2 and TNF-α co-immunostaining was performed in non-tumor colon sections from non-treated and from OxPt-treated patients, (Figure 5).

A low intensity level of TNF-α staining was found in samples from untreated patients, with only a few cells showing medium TNF-α staining intensity colocalizing with an AmotL2 signal of medium intensity level (Figure 5, panels A–D). Co-immunostaining for AmotL2 and TNF-α at medium to high intensity in both glandular and ENS cells was found in healthy colon sections from patients who underwent OxPt therapy (Figure 5, panels E–H).

### 3.5. Variations in AmotL2 Expression in the Enteric Nervous System in Samples from Patients Treated with OxPt-Based Chemotherapy

The expression of AmotL2 in ENS cells was evidenced by co-immunostaining with MAP2 (Figure 6) and GFAP (Figure 7) markers in non-tumor colon sections from untreated and OxPt-treated patients.

A low staining intensity for AmotL2 was found in neurons and glia in samples from untreated patients (Figure 6 and Figure 7, panels A–D). However, in samples from patients undergoing OxPt therapy, AmotL2 staining intensity was much higher than in those from non-treated patients in both GFAP- and MAP2-positive cells (Figure 6 and Figure 7, panels E–H).

Figure 8 displays the compilation of results for AmotL2, inflammation (TNF-α), and GFAP expression in the ENS in samples from patients suffering from CRC previously and after undergoing OxPt-based chemotherapy.

## 4. Discussion

Gastrointestinal dysfunction associated with many pathological conditions has been related to ENS damage [25] and, furthermore, to the non-desirable effects of OxPt-based chemotherapy, including GI neuropathy [26,27,28]. The ENS comprises both neurons that regulate contractility [23] and glial cells that support neurons and regulate its activity but is also related to intestinal epithelia barrier (IEB) maintenance [29] or epithelial cell proliferation and differentiation [30].

The present study was designed looking for insight into the pathogenesis of chemotherapy-induced peripheral neuropathy (CIPN) in humans. Since the introduction of OxPt-based chemotherapy in the neo-adjuvant therapy of colorectal cancer, colorectal resections have been a way to obtain enteral–neural tissue samples from OxPt-treated patients. However, in current databases, to our knowledge, there are no studies in humans reporting the effects of OxPt-based chemotherapy on the ENS.

Our results in humans show a decrease in the number of ENS neurons and ENS glial cells (EGCs) (Figure 2) following OxPt-based chemotherapy, in agreement with previous studies performed in mice [9], mostly those expressing nitric oxide (NO) synthase [27]. In mice, OxPt treatment induced a significant reduction in the total number and proportion of ChAT-IR neurons, with no significant differences in VAChT-IR nerve fiber density observed in the myenteric plexus of the colon following oxaliplatin treatment when compared with the vehicle-treated cohort [9].

Interestingly, we observed ENS cells in which both markers MAP2 and GFAP were positive in CRC patients (Figure 2). Previous studies established the potential of EGCs to acquire the properties of neural crest stem cells (NCSCs), given the fact that during development, precursor cells expressing GFAP are capable of generating different cell types in the central nervous system (CNS) [31,32,33]. Furthermore, some lineages of peripheral glial cells in adults have the ability to acquire neural stem cell properties, generating limited neurogenesis in response to injury or stress [34,35,36].

We found an increase in TNF-α expression, an early marker of inflammation, in ENS samples from patients treated with FOLFOX chemotherapy (Figure 4), while it was absent or very low in the non-OxPt-treated group (Figure 3). Our data show TNF-α expression prevalent in EGCs and scarce in neurons (Figure 4). This further confirms in humans the involvement of glial cells in the ENS’s inflammatory response to OxPt/leucovorin/5-FU-chemotherapy. Chemotherapy induces inflammation in target tissues, setting the basis for the concurrent use of chemotherapy and immune-checkpoint inhibitors [37]; furthermore, chemotherapy-induced inflammation contributes to therapeutic effects but also to adverse effects [38]. OxPt, along with DNA damaging effects on neural cells, triggers an inflammatory response in the ENS, playing an important role in the pathogenesis of ENS damage [39,40]. Specifically, EGCs have been previously implicated in pathological processes such as inflammation [41,42], promoting proinflammatory and anti-inflammatory effects. It has been suggested that EGC prevents gut inflammation due to the loss of enteric glia, which are associated with severe inflammation of the small intestine [43]. However, these cells might produce neuroinflammation and cell death by increasing NO and ATP release [44]. It is interesting to highlight that the patients in this study underwent surgery at least 4 weeks after the end of treatment, when the clinical toxicity on the fast-growing cells of the bone marrow and the GI mucosa was mostly reversed.

As patients undergo surgery at least one month after the last chemotherapy, the presence of inflammation-related EGCs after recovery of the acute toxicity suggests that these cells are involved in the long-lasting GI toxicity of OxPt-based chemotherapy. The presence of diarrhea or constipation in about 49% of patients, years after completion of chemotherapy [6], indicates a long period of time with toxicity and that other parts of the bowel must be affected. Studies should be directed to establish if this long-lasting inflammation of the mucosa results from the direct effects of chemotherapy or if it includes more complex effects from chemotherapy-related changes in the colonic microbiota. As our samples only show changes in the colon, they need to be completed with those effects of OxPt-based chemotherapy on the stomach, complementary to those regarding physical exercise and modulation of the purinergic system in cisplatin-induced GI dysmotility [45] and the small bowel.

The parallel increase in inflammation marker TNF-α, along with AmotL2 expression in EGCs, varying from low TNF-α/medium AmotL2 signal in controls to high TNF-α/high AmotL2 after treatment (Figure 5), are interesting evidence that might indicate the involvement of EGCs in inflammatory bowel disease [46]. Our results indicate a relationship between TNF-α and AmotL2 expression, either in healthy areas of the colon from patients suffering from colon cancer resected previous to OxPt chemotherapy or in healthy areas of resected colon from patients after chemotherapy. However, TNF-α- and AmotL2-specific fluorescence is higher in samples taken after chemotherapy. These facts show a “basal” inflammatory state of the ENS accompanied with an elevation of AmotL2 in colon cancer and an increased inflammatory state with OxPt-based chemotherapy (Figure 5).

A previous study of our group reported changes in AmotL2 expression in colon cancer and its liver metastases after treatment with OxPt-based chemotherapy [14]. These changes were predominantly in cells of the immune system, neurons, and glial cells [14]. AmotL2 regulates many physiological and pathological processes like angiogenesis, cell polarity, cell proliferation and migration, and epithelial–mesenchymal transition [20,47]. Further studies must determine if the chemotherapy-related increase in the expression of AmotL2 and TNF-α are related or independent events. TNF-α is involved in the pathophysiology of inflammatory bowel disease, contributing, together with IFN-γ, to epithelial cell apoptosis [48]. The coincident high expression of AmotL2 in the epithelial cells could be explained by its antiapoptotic actions or its involvement in cell junctions and in the preservation of cell polarity in response to the mucosal toxicity of chemotherapy [47]

In the present study we found a low expression of AmotL2 in the neurons and glia of the ENS of untreated patients (Figure 6 and Figure 7: panels A–D), while the treated group showed a high expression of AmotL2 (Figure 6 and Figure 7: panels E–H). This higher expression of AmotL2 in the enteric neurons of the treated patients could reflect its involvement in the acute and reparative process of the OxPt-induced neuronal damage. This fact is reinforced by the finding in cultured neurons demonstrating the role of AmotL2 in the development of the dendritic tree organization [20] and in the synaptic maturation podosome formation [22]. Furthermore, AmotL2 is an inhibitor of the intracellular Hippo–Yap pathway, which is involved in organ formation during embryogenesis [15] and regulates some processes in the main glial cells of the peripheral nervous system, which are the Schwann cells [49]. Injury to the Schwann cells will produce an activation of the Hippo–Yap pathway, resulting in cell proliferation, migration, and differentiation [50]. This activation of the Hippo–Yap pathway will also implicate an overexpression of its regulator AmotL2; consequently, Amot L2 seems to be upregulated in order to lower inflammation response to OxPt toxicity and, through the activation of Hippo–Yap signaling, prevent fibrosis, as similarly reported for non-alcoholic steatohepatitis in a rat model [51].

Altogether, our study in humans demonstrates the involvement of AmotL2 in the response of RNS glial cells to OxPt-based chemotherapy. However, as few studies of AmotL2 have been performed, more extended research, including on the vascular–capillary–endothelial aspects and the microenvironment, needs to be performed in order to state if this is a circumstantial fact or a reaction against inflammation. Regarding regeneration after toxicity, AmotL2 could play a pivotal role during differentiation of pluripotent stem cells, as recently pointed out by the authors of [21]. The strengths and limitations of this study come from the same source, the human origin of the sample. It has the value of real-life studies, but, despite the selection, there is variability in the homology and quality of the samples. The combination of human studies with rodent models will undoubtedly contribute to better understanding the involvement of AmotL2 in OxPt toxicity, inflammation, and peripheral neuron regeneration.

## Figures and Tables

**Figure 1 biomedicines-12-01952-f001:**
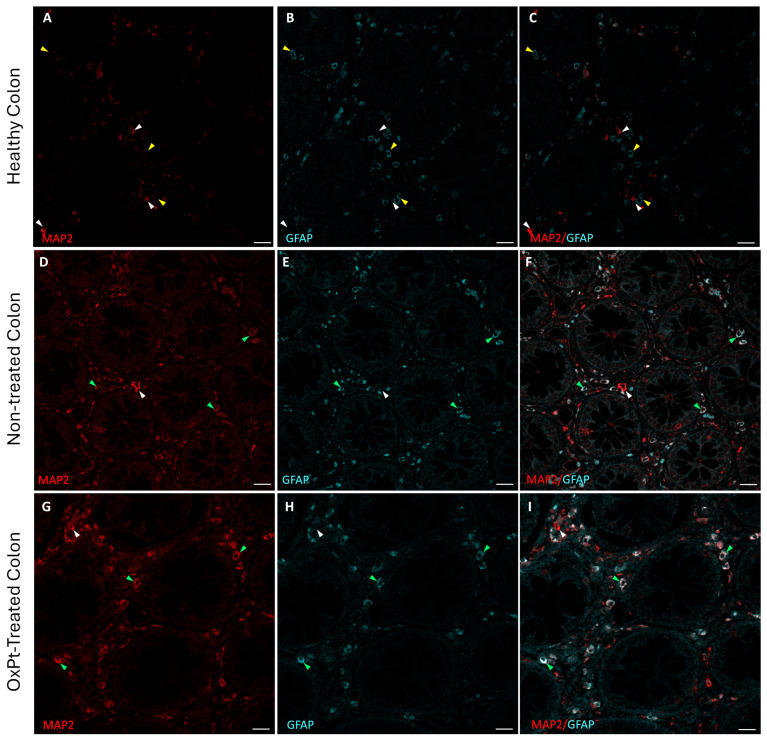
Double staining for MAP2 (red) and GFAP (cyan) in human enteral nervous system. Panels (**A**–**C**): MAP2+ cells (white arrowheads) and GFAP+ cells (yellow arrowheads) in control with healthy colon. Panels (**D**–**F**): healthy colon areas from not-yet-treated CRC patients; most of the cells show both MAP2+ and GFAP+ staining (green arrowheads) and a few MAP2+ cells and GFAP− cells (white arrowheads). Panels (**G**–**I**): samples from OxPt-treated CRC patients; MAP2+ and GFAP+ staining (green arrowheads), a few MAP2+/GFAP− cells (white arrowheads) or GFAP+/MAP2- cells (yellow arrowheads). Bar = 20 µm.

**Figure 2 biomedicines-12-01952-f002:**
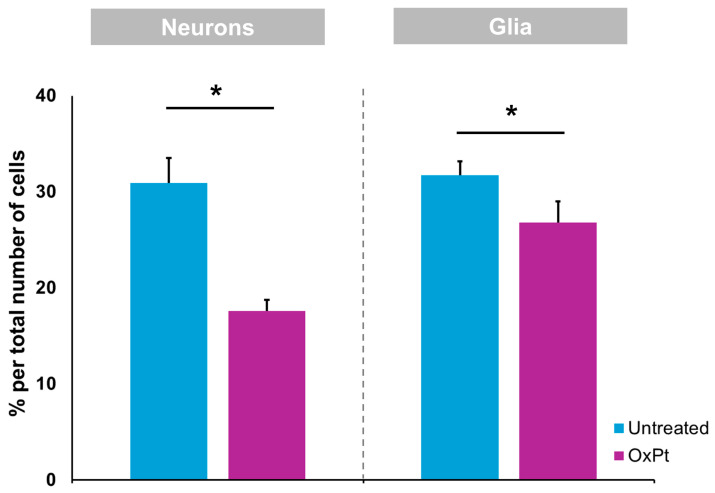
Oxaliplatin treatment induced a reduction in MAP2- and GFAP-immunoreactive cells within the colon myenteric plexus. Wholemount preparations of the samples containing colon myenteric plexus were labeled with MAP2 and GFAP. Oxaliplatin treatment induced a significant reduction in the density of MAP2- and GFAP-immunoreactive cells within the myenteric plexus when compared with samples obtained previous to OxPt-treated group. Untreated = samples from patients previous to chemotherapy. OxPt = samples from patients after chemotherapy. The data are expressed as the mean ± SD. * *p* < 0.01; *n* = 5/group.

**Figure 3 biomedicines-12-01952-f003:**
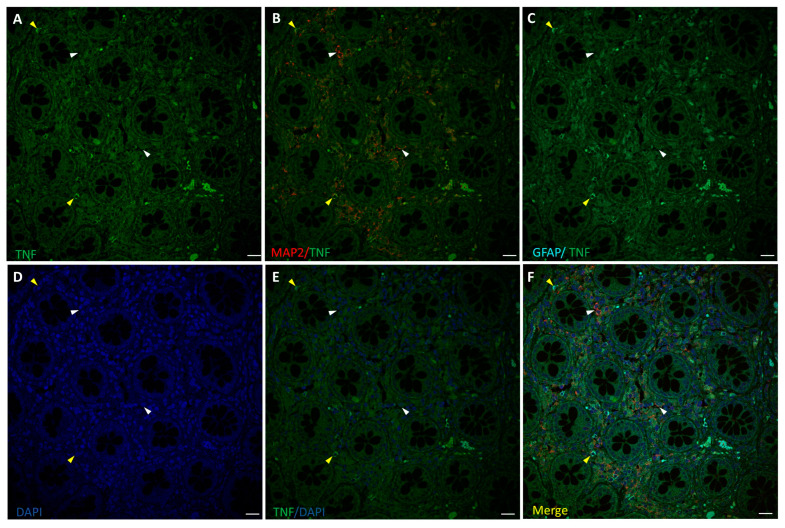
Staining for TNF-α (green), MAP2 (red), and GFAP (cyan) in human non-tumor (healthy) colon from untreated CRC patients. Panels (**A**–**C**): TNF-α staining, MAP2+ (white arrow-heads), and GFAP+ (yellow arrowheads) of ENS surrounding colon crypts. Panels (**D**–**F**): DAPI, DAPI/TNF-α staining, and merge of DAPI/TNF-α/MAP2/GFAP. Specific fluorescence for TNF-α was positive only in a few GFAP+ cells (**C**). Bar = 20 µm.

**Figure 4 biomedicines-12-01952-f004:**
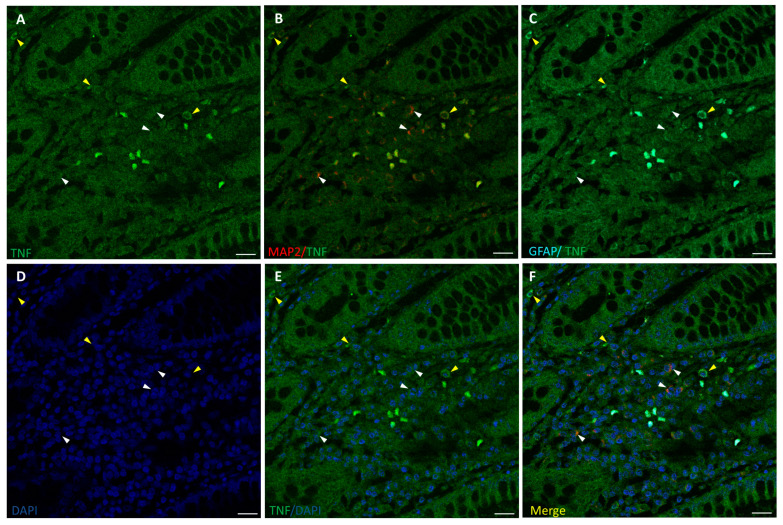
Staining for TNF-α (green), MAP2 (red), and GFAP (cyan) in human non-tumor (healthy) colon from OxPt-treated CRC patients. Panels (**A**–**C**): TNFα+/MAP2+ cells (white arrows) and TNFα+/GFAP+ cells (yellow arrows). Panels (**D**–**F**): DAPI, DAPI/TNF-α staining, and merge of DAPI/TNF-α/MAP2/GFAP. Bar = 20 µm.

**Figure 5 biomedicines-12-01952-f005:**
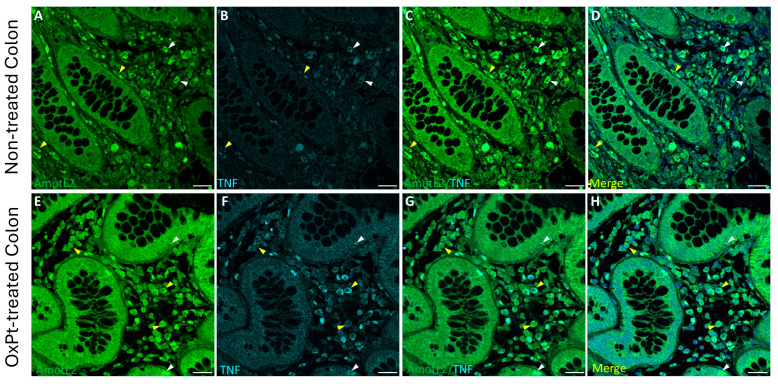
Double staining of AmotL2 (green) and TNF-α (cyan) in non-tumor colon areas from untreated and OxPt-treated CRC patients. Panels (**A**–**D**): AmotL2+/TNF-α colocalized staining at variable intensities, medium (white arrowheads) or low (yellow arrowheads); AmotL2+/TNF-α/DAPI merge. Panels (**E**–**H**): AmotL2+ at high staining level (yellow arrowheads) and TNF-α+ colocalized in some cells (white arrowheads). TNF-α signal intensity varied parallel to that of AmotL2. Panels D and H show AmotL2/TNF-α/DAPI merges. Bar = 20 µm.

**Figure 6 biomedicines-12-01952-f006:**
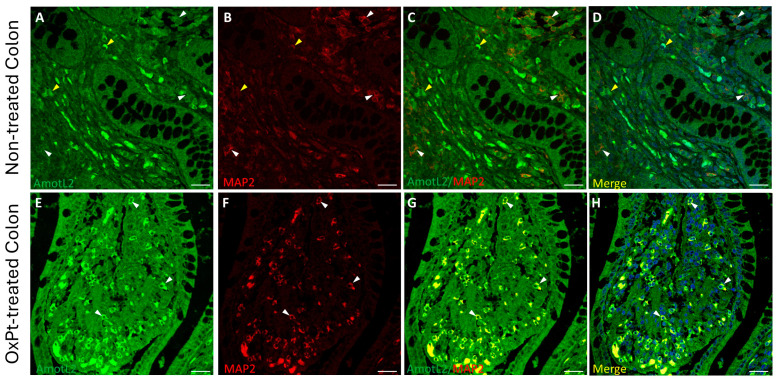
Double staining of AmotL2 (green) and MAP2 (red) in non-tumor colon areas from untreated and OxPt-treated CRC patients. Panels (**A**–**D**): AmotL2+ fluorescence staining at low/medium intensity in MAP2+ cells (white arrowheads) and AmotL2+ fluorescence staining at medium intensity in MAP2- cells (yellow arrowheads). Panels (**E**–**H**): high AmotL2 staining intensity in MAP2+ cells (white arrowheads). Bar = 20 µm.

**Figure 7 biomedicines-12-01952-f007:**
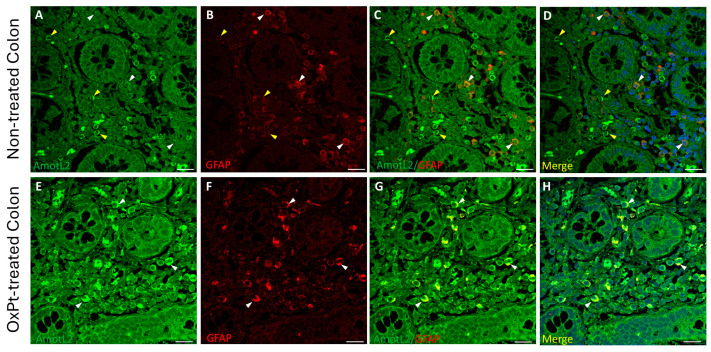
Double staining of AmotL2 (green) and GFAP (red) in non-tumor colon areas from untreated and OxPt-treated CRC patients. Panels (**A**–**D**): low AmotL2+ fluorescence signal in GFAP+ cells (white arrowheads). Some AmotL2+/GFAP− cells (yellow arrowheads). Panels (**E**–**H**): AmotL2+ staining in GFAP+ cells (white arrowheads). Bar = 20 µm.

**Figure 8 biomedicines-12-01952-f008:**
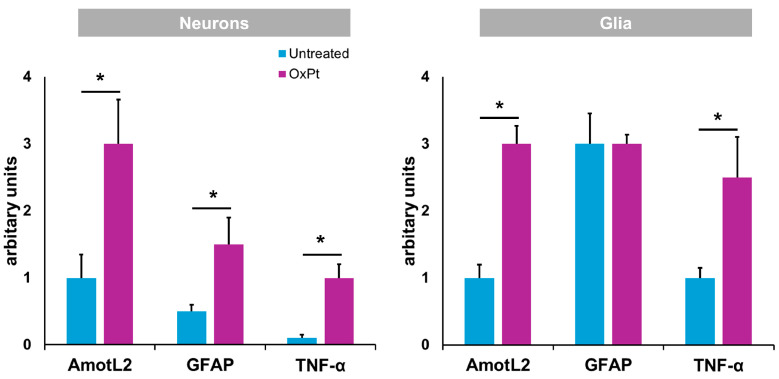
Specific fluorescence signal for AmotL2, GFAP, and TNF-α in neurons and glia from untreated and OxPt-treated CRC patients. Untreated = samples from patients previous to chemotherapy. OxPt = samples from patients after chemotherapy. The data are expressed as the mean ± SD. * *p* < 0.01.

## Data Availability

The data presented in this study are available on request from the corresponding author.

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
