# Peer review of "Changes in AmotL2 Expression in Cells of the Human Enteral Nervous System in Oxaliplatin-Induced Enteric Neuropathy"

_biomedicines, 2024, doi:10.3390/biomedicines12091952_

Round 1
Reviewer 1 Report (Previous Reviewer 1)
Comments and Suggestions for Authors
The author has made the changes I requested. It is acceptable to receive this manuscript.
Reviewer 2 Report (Previous Reviewer 3)
Comments and Suggestions for Authors
Dear authors
I appreciate your reply to my questions. This new version is better and clear. I don't have any questions. My asks were response.
Congrats.
This manuscript is a resubmission of an earlier submission. The following is a list of the peer review reports and author responses from that submission.
Round 1
Reviewer 1 Report
Comments and Suggestions for Authors
This article examined changes in AmotL2 expression in human enteric nervous system cells in OxPt-induced enteric neuropathy. It was found that in colorectal cancer patients treated with oxidized platinum chemotherapy, AmotL2 expression was increased in the enteric nervous system cells of untreated and oxidized platinum-treated patients and correlated with inflammation. This suggests that AmotL2 plays a role in oxidized platinum-induced enteric nervous system toxicity. But I have a couple questions.
1. How do changes in AmotL2 expression affect cells of the human enteric nervous system in oxaliplatin-induced enteric neuropathy? This should be added in the INTRODUCTION section.
2. Is the high expression of AmotL2 in the treated group related to the acute and reparative processes of oxaliplatin-induced nerve injury? Is AmotL2 involved in the inflammatory response to oxaliplatin-induced enteric nerve injury? Should be added to the discussion.
3.The Figure Bar values of the manuscript are very unclear and need to be revised.
4.Is there a somewhat long difference between the time of funding of the author's project and the time of the ethical certification (2020 and 2016). This issue needs to be explained.
Reviewer 2 Report
Comments and Suggestions for Authors
The authors performed immunohistochemistry of oxaliplatin (OxPt)-treated colorectal cancer (CRC) patients. Although the study using human samples is valuable, this manuscript has a serious flaw that it lacks quantitative analysis. For this reason, this manuscript is difficult to be published in Biomedicines.
1. As mentioned above, all of the immunohistochemical results should be quantified. Otherwise, conclusions cannot be convinced. Tables 1 and 2 are too vague and appear subjective.
2. Seeing Table 2, AmotL2 expression appears not to be so different between untreated and OxPt-treated patients. This impression conflicts with "AmotL2 differential expression". Quantitative analysis might resolve this conflict.
3. In Abstract, it is written that "the total amount of neurons and glia cells decreased in OxPt-treated patients" (lines 31-32). Which data is this conclusion based on?
4. Figures 1-3 are too dark and low-contrast except TNF-α signals in Figure 3.
5. To show colocalization in Figures 2-6, high-power view images are needed.
6. The sample in Figure 1A-C is a healthy colon from a non-patient? Please make clear.
Comments on the Quality of English LanguageModerate editing of English language is required.
Reviewer 3 Report
Comments and Suggestions for Authors
The study Changes in AmotL2 Expression in Cells of the Human Enteral 2 Nervous System in Oxaliplatin-Induced Enteric Neuropathy is important and relevant due to the topic related to the ENS.
It is already well described that the use of OXP significantly modifies GI functions and that this effect may have a direct relationship with changes in vagal autonomic pathways and physical exercise.
DOI. 10.1016/j.lfs.2020.118972
DOI. 10.1590/1414-431X2024e13234
In this sense, some points must be clarified.
1. How are the cholinergic neurons in these patients?
2. Images could be of better quality. Furthermore, the comparisons between the "n" are not clear. Clearer graphs of neuronal values should be presented in order to better understand possible differences.
3. Why was only TNF-alpha evaluated and not other inflammatory cytokines more focused on changes such as IL-33 and IL-17?
4. The discussion is too short without making any progress. I suggest a deeper discussion. Furthermore, it is important to present the limitations of the study and future perspectives.